# Dengue Seroprevalence and Seroconversion in Urban and Rural Populations in Northeastern Thailand and Southern Laos

**DOI:** 10.3390/ijerph17239134

**Published:** 2020-12-07

**Authors:** Dyna Doum, Hans J. Overgaard, Mayfong Mayxay, Sutas Suttiprapa, Prasert Saichua, Tipaya Ekalaksananan, Panwad Tongchai, Md. Siddikur Rahman, Ubydul Haque, Sysavanh Phommachanh, Tiengkham Pongvongsa, Joacim Rocklöv, Richard Paul, Chamsai Pientong

**Affiliations:** 1Tropical Medicine Graduate Program, Academic Affairs, Faculty of Medicine, Khon Kaen University, Khon Kaen 40002, Thailand; dynadoum@gmail.com (D.D.); sutasu@kku.ac.th (S.S.); prasertsai@kku.ac.th (P.S.); 2Department of Microbiology, Faculty of Medicine, Khon Kaen University, Khon Kaen 40002, Thailand; hans.overgaard@nmbu.no (H.J.O.); tipeka@kku.ac.th (T.E.); panwad1622@gmail.com (P.T.); siddikur@brur.ac.bd (M.S.R.); 3Faculty of Science and Technology, Norwegian University of Life Sciences, P.O. Box 5003, 1432 Ås, Norway; 4Institute of Research and Education Development (IRED), University of Health Sciences, Ministry of Health, P.O. Box 7444, Vientiane 43130, Laos; mayfong@tropmedres.ac (M.M.); sysavanhp@gmail.com (S.P.); 5Lao-Oxford-Mahosot Hospital-Wellcome Trust Research Unit (LOMWRU), Mahosot Hospital, Vientiane 43130, Laos; 6Centre for Tropical Medicine and Global Health, Nuffield Department of Clinical Medicine, Old Road Campus, University of Oxford, Oxford OX3 7LG, UK; 7HPV & EBV and Carcinogenesis Research Group, Khon Kaen University, Khon Kaen 40002, Thailand; 8Department of Statistics, Begum Rokeya University, Rangpur 5400, Bangladesh; 9Department of Biostatistics and Epidemiology, University of North Texas Health Science Center, Fort Worth, TX 76177, USA; mdubydul.haque@unthsc.edu; 10Savannakhet Provincial Health Department, Savannakhet 13000, Laos; tiengkhampvs@gmail.com; 11Department of Public Health and Clinical Medicine, Umeå University, 90187 Umeå, Sweden; joacim.rocklov@umu.se; 12Unité de la Génétique Fonctionnelle des Maladies Infectieuses, Institut Pasteur, CNRS UMR 2000, 75015 Paris, France

**Keywords:** dengue, DENV, seroprevalence, seroconversion, socioeconomic factors

## Abstract

Dengue is the most rapidly spreading mosquito-borne viral disease in the world. The detection of clinical cases enables us to measure the incidence of dengue infection, whereas serological surveys give insights into the prevalence of infection. This study aimed to determine dengue seroprevalence and seroconversion rates in northeastern Thailand and southern Laos and to assess any association of mosquito control methods and socioeconomic factors with dengue virus (DENV) infection. Cross-sectional seroprevalence surveys were performed in May and November 2019 on the same individuals. Blood samples were collected from one adult and one child, when possible, in each of 720 randomly selected households from two urban and two rural sites in both northeastern Thailand and southern Laos. IgG antibodies against DENV were detected in serum using a commercial enzyme-linked immunosorbent assay (ELISA) kit. Overall, 1071 individuals participated in the study. The seroprevalence rate was high (91.5%) across all 8 study sites. Only age and province were associated with seroprevalence rates. There were 33 seroconversions during the period from May to November, of which seven reported fever. More than half of the seroconversions occurred in the rural areas and in Laos. Dengue seroconversion was significantly associated with young age (<15 years old), female gender, province, and duration of living in the current residence. No socioeconomic factors or mosquito control methods were found to be associated with seroprevalence or seroconversion. Notably, however, the province with most seroconversions had lower diurnal temperature ranges than elsewhere. In conclusion, our study has highlighted the homogeneity of dengue exposure across a wide range of settings and most notably those from rural and urban areas. Dengue can no longer be considered to be solely an urban disease nor necessarily one linked to poverty.

## 1. Introduction

Dengue is the most rapidly spreading arboviral disease in the world [1] and has become a leading cause of illness and death in the tropics and subtropics [2], with annual mortality figures of > 20,000 deaths reported [3]. The incidence of dengue has been dramatically increasing in the past two decades, with the number of symptomatic dengue infections more than doubling every ten years [4]. Currently an estimated 390 million people are infected per year of which 96 million are clinically apparent [5]. This prolific increase has, at least in part, been exacerbated by urban development, an environment to which the major mosquito vector species, *Aedes aegypti*, is well-adapted. Numerous studies have reported on the association of dengue disease with socioeconomic, demographic and infrastructure features [6]. Macro-level risk factors including overpopulation, uncontrolled urbanization and poor waste management play prominent roles in the emergence of dengue [7]. Environmental factors, global trade and travel, climate variability and inadequate vector control influence global dengue transmission and persistence [8]. The role of socioeconomic conditions are increasingly recognized as potentially important for the local risk of dengue virus (DENV) infection, with worsening socioeconomic conditions and lack of knowledge enhancing the risk of exposure [9]. However, the extent to which low socioeconomic status is linked to dengue incidence is highly variable and may be location-specific [10]. This will require identifying those factors of local importance to aid the development of targeted interventions for reducing the burden of dengue. Current strategies to mitigate the burden are based solely on mosquito control through national source reduction campaigns and fogging in and around dengue case premises. Community level efforts in environmental management via cleaning campaigns have been shown to have significant impacts on dengue transmission [11]. In addition, there is an increasing use of household level protection methods, such as window screens, mosquito nets, coils, sprays and larviciding of water containers, but which are of uncertain efficacy [12].

Whilst improved diagnostics are enabling a more accurate measure of the incidence of disease, given the large but variable proportion of infections that are sub-clinical or asymptomatic, serological studies provide a better measure of the true force of infection for risk factor analysis [13,14]. The objectives of this study were to: (i) determine dengue seroprevalence rates, (ii) determine seroconversion rates (i.e., sero-incidence) over a 6-month period, (iii) estimate the proportion of asymptomatic infections and (iv) assess the association of socioeconomic factors and mosquito prevention methods with the dengue seroprevalence and seroconversion rates.

## 2. Materials and Methods

### 2.1. Study Design

This study was part of a larger project on dengue risk factors, vulnerability, and climate conducted in representative urban and rural areas near or along the Mekong River in Mukdahan and Ubon Ratchathani provinces in northeastern Thailand and in Savannakhet and Champasak provinces in southern Laos (Figure 1).

In each of the four provinces, one urban and one rural study site were selected based on high incidence of dengue during 2013–2017. Urban and rural areas were defined according to the country-specific classification of rural and urban areas [15]. In addition, rural areas were determined as areas outside an urban setting and where agriculture was the main activity [16]. We aimed to recruit 180 individuals from each site and, assuming a household size of four persons, about 90 households (HHs) were randomly selected giving a total of 720 HHs. Two cross-sectional seroprevalence surveys were carried out before and after the rainy season, the first in May 2019 and the second in November 2019 in each of the selected households in the eight sites, to assess previous exposure to dengue and seroconversion rates.

Additional individual information related to potential dengue risk factors, such as patient’s age, gender, vaccination history, and any memory of previous dengue infection were collected during the blood sample collections. Household information was collected at the beginning of the project during September 2018–February 2019, including the number of people living in the household, use of any kind of larval and adult mosquito control, whether sleeping under bed net (day vs. night) and presence of window screens and specific features used to determine socioeconomic status (see below). In addition to the cross-sectional surveys, a fever survey was carried out throughout the study period with weekly visits to participating HHs and participants asked as to whether they had any fever episodes during the preceding week. Body temperature using an axilla thermometer was measured in subjects who reported a recent or current fever.

### 2.2. Blood Sample Collection

In May 2019, two randomly selected household participants (one adult ≥18 years old and one child ≥5 years old, if available) from each household were asked for consent to participate in the seroprevalence study. A blood sample of 3 mL venous blood was taken into ethylenediamine tetraacetic acid (EDTA)-tubes from each participant. Serum was separated by centrifugation at 1300 ×g for 10 min, kept at + 4 °C, and transported to the laboratory of Khon Kaen University (KKU) within two days for further processing. This process was repeated in the same households in November 2019.

### 2.3. Serological Analysis

DENV anti-IgG antibody detection in the collected sera was performed using a commercial enzyme-linked immunosorbent assay (ELISA) (catalog numbers EI 266a-9601-1 G, EUROIMMUN, Luebeck, Germany) following the manufacturer’s instructions. The kit’s specificity and sensitivity are 0.988 (95% CI: 0.979–0.993) and 0.892 (95% CI: 0.879–0.903), respectively [17]. Briefly, serum samples diluted 1:101 were added to microplates and optical densities (OD) measured with a microplate reader. Antibody values ≥20 relative units/mL were considered seropositive. Seroconversion was defined as a four-fold increase in the IgG antibody titer in paired samples from May and November.

### 2.4. Meteorological Variables

Continuous temperature, humidity and rainfall data were collected using on-site weather stations (Davis Vantage Pro2, Davis Instruments Corporation, Davis, CA, USA), except in That Kaen, Mukdahan, Thailand where data were retrieved from the provincial Department of Meteorology. The monthly mean of the daily humidity, maximum, minimum and mean temperatures were then calculated. The monthly mean of the diurnal temperature range (DTR) was also calculated. Precipitation was cumulated over the month.

### 2.5. Data Analysis

The socioeconomic status (SES) of each household was calculated using Principal Components Analysis (PCA) based on group-weighted mean scores on a range of household variables and items [18,19]. The items used in the wealth status ranking included House roof material (Ceramic, Wood, Metal); House walls material (Plastered, Cement, Bricks, Wood); household ownership of durable consumer assets (Television, VCD, Refrigerator, Washing machine, Mobile Phone, Smartphone, Computer, Oven, Microwave, Air conditioner, Personal Car, Pickup, Motorcycle); Toilet facility; The material of the Toilet/bathroom floor (Tiles, Cement, Earth); and Flush toilet/squat toilet. A descriptive analysis of the different asset variables was carried out to determine their frequency and standard deviation. A co-variance matrix was generated for the PCA analysis as all the variables were standardized to the same unit (binary yes = 1/no = 0). These results were used to create a wealth score by ranking each household into low, intermediate, or high SES groups at the 33rd and 66th percentiles (tertiles).

Analysis of risk factors associated with DENV IgG antibody seroprevalence (May survey) was performed by fitting Generalized Linear Mixed Models (GLMM) with binomial error structure (i.e., a logistic regression). Forty-five individuals confirmed having had the dengue vaccine and 51 individuals did not know whether they had had the dengue vaccine. Both these groups of individuals were excluded from the seroprevalence statistical analyses. All factors such as age group, gender, previous exposure to dengue, yellow fever immunization, province, setting (urban vs. rural), country, duration of living in residence, house density, number of people living in household, socioeconomic status, any kind of larval control (temephos and cleaning containers), any kind of adult mosquito control (mosquito nets, household insecticide sprays, fogging, mosquito swatter, and repellent), sleeping under bed net (day vs. night) and window screen were included in the univariable analysis. Household was fitted as the random factor to take into account repeated sampling (more than one individual) from the same household. A GLMM was also fitted to analyze the same risk factors associated with seroconversion. Variables that were associated with DENV IgG seroprevalence (or seroconversion) with a *p*-value of < 0.25 were subsequently included in the multivariable model. Statistical Package for Social Science version 23 (SPSS, Inc., Chicago, IL, USA) and GenStat version 15 (VSN International Ltd, Hemel Hempstead, UK) was used for all data analyses [20]. To explore differences in meteorological variables (mean monthly minimum, maximum and average temperatures, mean monthly diurnal temperature range, mean monthly humidity and cumulative monthly rainfall) amongst provinces, we fitted a GLM with normal error structure. An overdispersion parameter was estimated to take into account any overdispersion of the data.

### 2.6. Ethical Considerations

This study was approved by the Khon Kaen University Ethics Committee for Human Research (ref. no. HE611228, 02/08/2018 and HE631006, 29/01/2020), National Ethics Committee for Health Research of Laos (ref.no. 057/NECHR, 15/05/2018) and the Regional Committees for Medical and Health Research Ethics in Norway (2018/1085/REK sør-øst C, 27/06/2018). The purpose of the study was explained to all participants and written informed consent/assent was obtained from study participants. Participants ≥18 years old provided signed informed consent. Adolescents aged 13–17 years provided a signed informed consent form together with their parent/guardian. Children aged 7–12 years provided signed informed assent and their parent/guardian provided signed informed consent. A parent/guardian provided signed consent for children aged 5–6 years. All the data and samples were analyzed without name, ID card number, or other directly recognizable types of information.

## 3. Results

### 3.1. Characteristics of the Study Population

A total of 1071 participants were recruited into the study of which 304 (28.4%) were from Mukdahan province, Thailand 290 (27.1%) from Ubon Ratchathani province, Thailand, 241 (22.5%) from Savannakhet province, Laos and 236 (22%) from Champasak, Laos. The number of participants from rural areas (n = 562) and urban areas (n = 509) were similar (52% vs. 48%). The age range was 5 to 93 years and mean age of all participants was 43.4 years (SD 20.5); two thirds of participants were female (66.3%). The wealth scoring classified 328 (30.6%) households as high SES, 365 (34.1%) as intermediate SES and 378 (35.3%) as low SES. There were more high SES households in urban areas (n = 229, 69.8%) than in rural areas (n = 99, 30.2%). In contrast, there were more low SES households in the rural areas (n = 266, 70.4%) compared to the urban areas (n = 112, 29.6%). The mean house density was higher in rural areas than in urban areas in all four provinces. The details of related information are shown in Table 1.

### 3.2. Dengue Antibody Seroprevalence in May 2019

Overall, 980/1071 (91.5%) of the serum samples collected in May 2019 were anti-DENV IgG antibody positive. The seropositivity in northeastern Thailand province was similar in both Mukdahan (90.5%) and Ubon Ratchathani (91.4%). The seropositivity in Champasak (95.3%) was higher than in Savannakhet (89.2%) in southern Laos. Seropositivity increased with age, with an approximate 20% increment every 5 years up to the age of 20. The seroprevalence was 100% for participants above the age of 25 years in Thailand and above the age of 43 years in Laos. The IgG OD with age was consistently higher in rural settings from both countries (Figure 2). There were 88.9% of male participants and 92.8% of female participants found to be dengue antibody seropositive. In Thailand, the seropositive rates in the rural settings were 88% (Mukdahan 85.3% vs. Ubon Ratchathani 90.9%) and in the urban settings 94.2% (Mukdahan 96.5% vs. Ubon Ratchathani 91.9%). In rural Laos the seropositive rate was 91.4% (Savannakhet 88.5% vs. Champasak 94.7%), while it was 93.1% in the urban settings (Savannakhet 90% vs. Champasak 95.9%) (Table 2). In 350 households both of the tested participants were seropositive. In 280 households one of the two tested participants was seropositive. In 11 households both tested participants were seronegative.

### 3.3. Association between Dengue Antibody Seroprevalence and Potential Risk Factors

In the univariable analyses, age group, duration of living in the residence (years), number of people living in the household, SES, any kind of larval mosquito control and window screen were significantly associated with positive DENV IgG seroprevalence in May 2019. As compared to children 5–9 years old, the odds of being seropositive increased with age: 10–14 years (Odds Ratio (OR) = 2.03, 95% Confidence Intervals (CI): 1.02–4.03), 15–19 years (OR = 8.22, 95% CI: 3.03–22.30), ≥20 years (OR = 180.74, 95% CI: 70.1–465.8). Participants living in their current setting for >5 years had a significantly higher odds of being seropositive in comparison to those residing there for less than five years (OR = 3.68, 95% CI: 1.61–8.42). Individuals living in households with more than five persons had lower odds of being seropositive than in households with 1–3 members (OR = 0.45, 95% CI: 0.23–0.87). Individuals living in households with low SES (OR = 0.53, 95% CI: 0.29–0.99) had significantly lower odds of being dengue seropositive than those living in high SES households. Households without larval control were associated with a lower seroprevalence compared to households with larval control (OR = 0.41, 95% CI: 0.21–0.83). Households with no window screen were associated with lower seroprevalence (OR = 0.26, 95% CI: 0.08–0.85) (Table 2). The participants in Champasak province had near significant higher odds (OR = 2.15, 95% CI: 0.99–4.66) of being seropositive than those in Mukdahan which was the reference province. Only three individuals had had a yellow fever virus vaccine and thus this variable was not analyzed.

In the multivariable analysis, only age group and province were found to be associated with seroprevalence, where all older age groups had significantly higher odds of being seropositive compared to the youngest age group (5–9 years) and participants in Champasak province had significantly higher odds of being seropositive compared to those in Mukdahan province.

### 3.4. Dengue Antibody Seroconversion

In the paired sera from May and November, the seroprevalence increased from 980 (91.5%) in May to 992 (92.6%) in November 2019. Ninety-two of the 1071 participants reported fever during the six-month period, of which 84 were seropositive (91.3%) and eight (8.7%) were seronegative. There were 33 seroconversions (3.1%) during the period, of which seven (21.2%) reported fever and 26 (78.8%) did not report fever. There were seven who seroconverted in Thailand of which two reported fever and 26 seroconversions in Laos of which five reported fever symptoms. Sixteen of the 33 seroconversions were converting from negative to positive, comprised of four individuals in the 5–9 years age group, nine in the 10–14 years age group, and three in the ≥20 years age group. Seventeen individuals who were seropositive in May had a four-fold increase in IgG titer in the November survey. These included five cases in the 5–9 years age group, four in the 10–14 years age group, two in the 15–19 years age group and six in the ≥20 years age group. Seroconversion was thus seen in all age groups: nine in the 5–9 years age group, 13 in the 10–14 years age group, two in the 15–19 years age group and nine in the ≥20 years age group. Of the 33 participants that seroconverted 18 were male and 15 females, 21 lived in rural areas and 12 lived in urban areas.

### 3.5. Association between Dengue Antibody Seroconversion and Potential Risk Factors

In the univariable analyses, age group, gender, country, province, duration of living in the residence (years), and any kind of larval mosquito control were significantly associated with DENV IgG seroconversion. As compared to children 5–9 years old, the odds of seroconverting were higher in the 10–14 years age group (OR = 4.80, 95% CI: 2.37–9.70), but lower in the older age groups (15–19 years OR = 0.25, 95% CI: 0.10–0.61 and ≥20 years OR = 0.06, 95% CI: 0.04–0.10). Female participants had lower odds of seroconverting than males (OR = 0.19, 95% CI: 0.13–0.26). Participants living in Laos had 5.5 times higher odds of seroconverting compared to Thailand. The participants in Champasak province had higher odds of seroconverting than those in Mukdahan (OR = 6.79, 95% CI: 1.98–23.35) which was the reference province. Participants living in their current setting for >5 years had a significantly lower odds of seroconverting (OR = 0.04, 95% CI: 0.02–0.10) compared to those living in their households for ≤5 years. Households without larval control were associated with a higher seroconversion probability compared to households with larval control (OR = 3.91, 95% CI: 1.10–13.89).

The final multivariable model of the GLMM identified several factors associated with dengue antibody seroconversion rate: age group, gender, province and duration of living in the residence. The odds of participants of age 10–14 years seroconverting during the observation period was 5.27 times higher than the odds of those from the 5–9 years age group. The odds of seroconversion in the 15–19 years (OR = 0.14, 95% CI: 0.05–0.37) and the ≥20 years (OR = 0.03, 95% CI: 0.02–0.06) age groups were lower than the 5–9 years reference group. The odds of female participants seroconverting were marginally higher than males (OR = 1.89, 95% CI: 1.14–3.11). The participants from Champasak province had much higher odds of seroconversion compared to Mukdahan province (OR = 39.89, 95% CI: 5.34–299.1). Participants living in their current setting for more than five years had significantly lower odds of seroconversion (OR = 0.005, 95% CI: 0.001–0.02) (Table 3). No socioeconomic, mosquito control or other variables were found to be associated with seroconversion.

To explore why Champasak had a much higher seroconversion rate than the other provinces, we assessed whether any of the meteorological variables were different among sites during the study period. Whilst cumulative monthly rainfall, and the minimum, mean and maximum temperatures were not different in Champasak compared to the other sites, both humidity and diurnal temperature range (DTR) were (Table 4). Humidity was higher in Ubon Ratchathani (t = 2.84, *p* = 0.007) and lower in Savannakhet (t = 2.07, *p* = 0.045). DTR was lower in Champasak than in any of the other provinces (Mukdahan t = 4.21, *p* < 0.001; Ubon Ratchathani t = 3.45, *p* = 0.001; Savannakhet t = 2.07, *p* = 0.045).

## 4. Discussion

To assess the potential risk factors associated with dengue virus seroprevalence rates among the urban and rural populations in northeastern Thailand and southern Laos, two cross sectional studies were performed over a six-month interval before and after the rainy season, in May and November 2019. The seroprevalence of DENV IgG antibodies was as high as 91.5 % across all the eight study populations. Such high seroprevalence rates have been previously observed in urban settings Malaysia (91.6%) [21], Bangladesh (80–93%) [7] and India (93%) [22] and are broadly comparable to those observed (79.2%) over a wider age range (6 months to 60 years) in a previous study in Thailand [23]. Nevertheless, finding such high seroprevalence rates across all eight sites covering rural and urban settings in two countries is surprising. Indeed, there was no difference in seroprevalence or seroconversion rates between rural and urban settings in any province from either country in multivariable analysis. Dengue has widely been accepted to be a predominantly contagious urban disease, because of the suitability of the urban environment for *Ae. aegypti*. However, DENV also circulates in rural areas and in addition to the rural vector species *Aedes albopictus*, *Ae. aegypti* also proliferates in rural environments, largely because of water storage practices [24]. There are an increasing number of studies worldwide highlighting the comparatively similar risk in rural and urban settings [21,25,26,27]. With developing economies, increasing urbanization and improved transport networks, rural areas are no longer as rural or as isolated as decades ago. Thus, viral circulation through population mobility and increasingly permissive environments for vector species likely contribute to increased homogeneity in dengue risk across diverse settings.

One notable observation was that the province consistently associated with the significantly highest seroprevalence and seroconversion rates also had the lowest Diurnal Temperature Range (DTR). DTR has been demonstrated to have effects on the biology of both the mosquito and the development of DENV within the mosquito, albeit at values of DTR much higher than observed here [28]. However, modelling studies have suggested than even in tropical regions with temperatures optimal for DENV transmission, even small variations in DTR can have an impact [29].

As expected, the seroprevalence rates increased rapidly with age, with as many as 41.2% of young children (aged 5–9) being seropositive and 100% by 25 years of age in Thailand and 43 years of age in Laos. By comparison, seroprevalence studies in central and southern Thailand found that 30.1% of children aged <9 years of age were seropositive [23]. Notably, there were increased odds of seroconversion in the 10–14 years age group as compared to the youngest age group, although the total number of seroconversions was small, abnegating any firm conclusions about increased risk. The very small number of seroconversions in the >20 years age group is likely to reflect the impact of immunity acquired to previous infection with homologous serotype(s), but which cannot be confirmed from the ELISA method used in our study. Of the 33 seroconversions, 16 were primary infections and 17 secondary or post-secondary infections. Seven of 33 seroconversions reported fever and 26 did not report fever. This confirms the increasing appreciation of the relative importance of inapparent vs. symptomatic infections and that secondary infections, although often leading to more severe disease, can be inapparent [30,31].

The association of gender with increased risk of infection is seemingly context-specific and likely to be dependent on differential exposure. Higher seroprevalence rates among females have previously been reported by others [32,33,34,35]. In many settings females spend more time at home and are thus more exposed to the peri-domestic mosquito population [32,36,37]. In contrast, a study in Singapore found that men were more likely to be exposed to mosquito bites than women during the daytime, at work or when traveling to/from work [38]. This suggests that more general outdoor work habits exposes males more than females [39].

Socioeconomic status has long been considered to be associated with risk of dengue, but the relationship is far from clear and varies according to the indicators used [40]. The poverty indicators being previously found to be most consistently associated with dengue risk are income and physical housing conditions [35,40]. We analyzed such indicators but found no association with risk of dengue. The lack of association of dengue with socioeconomic status in our study may be due to differences in the methodology of computing the SES score.

Vector control at the community or household level remains the only method to combat dengue. Whilst source reduction and community-based environmental management have shown some success [41,42], the efficacy of household level preventative measures is less clear. In a systematic meta-analysis only house window screening was found to have a protective effect against DENV infection [12]. Even a meta-analysis of the efficacy of the widely used larvicide temephos found highly variable efficacy in reducing mosquitoes and no evidence for having an impact on dengue incidence [43]. More recently, comparing dengue free vs. dengue case positive houses, a protective effect of indoor fumigation and mosquito nets was observed [44]. However, overall, the evidence that household level interventions reduce dengue incidence is largely absent. No household level preventative measures, either against adult or larval stage mosquitoes, were found to be associated with a reduction in seroprevalence or seroconversion in our study, consistent with another study [35].

The major strength of this study is the incorporation of eight study sites covering rural and urban settings in four different provinces from two different countries. There were, however, several limitations. Firstly, we did not specifically target all age groups. The protocol was to recruit a child and an adult from each participating household who were available at the time of the visit. This resulted in a very small number of children in the 5–9 year age group and a relatively small number of participants under the age of 20, beyond which age almost all participants were seropositive. This considerably reduced our power to detect significant associations of potential risk factors in the seroprevalence study. However, that the seroconversion associations also highlighted the same associations only with age and province lends support to our findings of no associations of dengue infection with socioeconomic status and mosquito prevention strategies. In addition, a similar study in Dhaka, Bangladesh found very similar seroprevalence and seroconversion rates to our study and did identify a protective effect of mosquito prevention tools [7]. The current study did not investigate the effects of participant’s dengue prevention related knowledge, attitude and practice (KAP) factors on their dengue seroprevalence. A study conducted in Malaysia suggested that KAP related factors are essential in potentially contributing to the development of a proactive program to protect the health of vulnerable groups in the communities. Proactive and sustainable efforts are needed to bring a behavioral change among communities in order to fight dengue outbreaks in endemic areas [45]. The higher numbers of females recruited was an unfortunate consequence of females (largely mothers) being more likely to participate in the study. A second limitation of our study was that the ELISA detection method could not differentiate between the different serotypes, how many previous infections an individual had, and thus whether individuals in the older age groups were immune to the viral serotypes circulating during the study. Nor can we be certain that seropositivity was not affected by cross-reaction with Zika or Japanese Encephalitis viruses (JEV) [46]. DENV and JEV are common endemic flaviviruses in Thailand and Laos. However, compared to dengue the prevalence of Japanese Encephalitis is relatively low. Zika is also present but with a low prevalence in Thailand and Laos. To date, there is no clear evidence to explain the relatively low prevalence of ZIKA virus infection in Asian countries [47]. Accordingly, our positive ELISA results are more likely to reflect dengue IgG than antibodies against other flaviviruses.

## 5. Conclusions

In conclusion, our study has highlighted the homogeneity of dengue exposure across a wide range of settings and most notably those from rural and urban areas. Dengue can no longer be considered to be solely an urban disease nor necessarily one linked to poverty. As is increasingly recognized, the incidence of inapparent infections outweighs that of symptomatic infections, thus underlining the importance of sero-surveys in risk factor analysis, rather than solely focusing on clinical cases.

## Figures and Tables

**Figure 1 ijerph-17-09134-f001:**
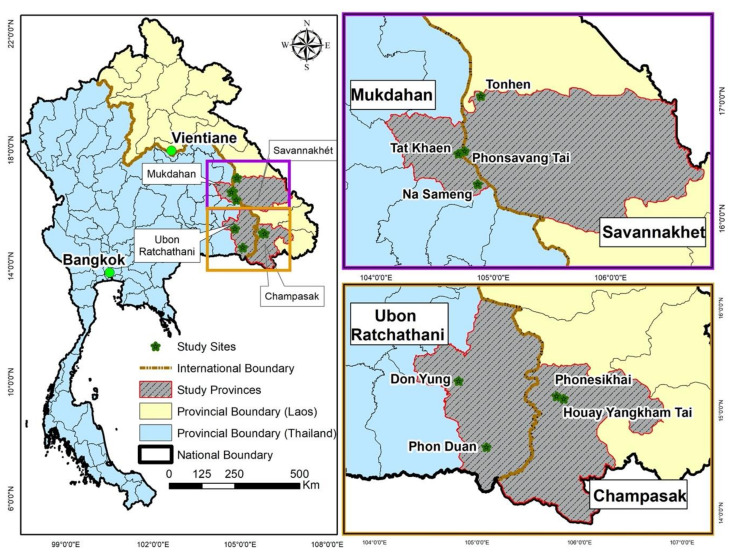
Study sites in Laos and Thailand.

**Figure 2 ijerph-17-09134-f002:**
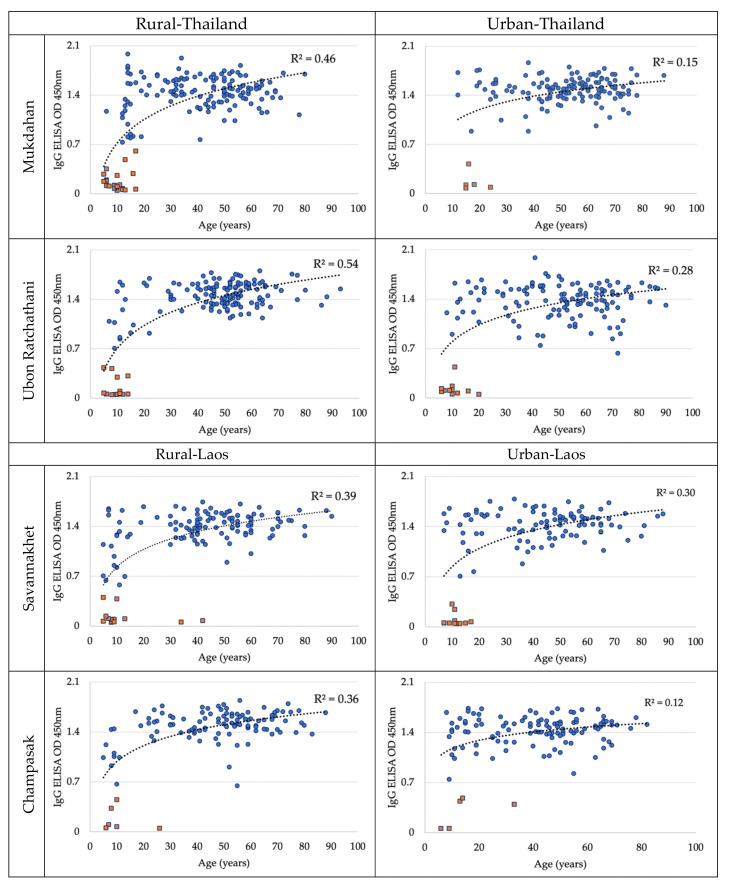
Distribution of dengue IgG enzyme-linked immunosorbent assay (ELISA) optical density (OD) values by age (blue dots denote the OD seropositive and orange dots denote the OD seronegative individuals) in urban and rural of Thailand (upper panels) and in Laos (lower panels) in May 2019. Shown is a logarithmic tendency curve.

**Table 1 ijerph-17-09134-t001:** General information of participants and households in Laos and Thailand in May 2019.

Variables		Thailand (n = 594)				Laos (n = 477)			
		Urban	Rural	Urban	Rural	Urban	Rural	Urban	Rural
**No. of individuals (%)**		141 (13.2)	163 (15.2)	136 (12.7)	154 (14.4)	110 (10.3)	131 (12.2)	122 (11.4)	114 (10.6)
**Age group (years)**									
	5–9	0 (0.0)	12 (7.4)	5 (3.7)	9 (5.8)	7 (6.4)	20 (15.3)	5 (4.1)	10 (8.8)
	10–14	2 (1.4)	24 (14.7)	11 (8.1)	15 (9.8)	10 (9.1)	11 (8.4)	11 (9.0)	4 (3.5)
	15–19	7 (5.0)	13 (8.0)	5 (3.7)	2 (1.3)	10 (9.1)	2 (1.5)	11 (9.0)	2 (1.7)
	≥20	132 (93.6)	114 (69.9)	115 (84.5)	128 (83.1)	83 (75.4)	98 (74.8)	95 (77.9)	98 (86.0)
**Gender**									
	Male	53 (37.6)	54 (33.1)	53 (39.0)	47 (30.5)	37 (33.6)	40 (30.5)	38 (31.2)	39 (34.2)
	Female	88 (62.4)	109 (66.9)	83 (61.0)	107 (69.5)	73 (66.4)	91 (69.5)	84 (68.8)	75 (65.8)
**Socioeconomic status**									
	High	70 (49.6)	20 (12.3)	59 (43.4)	28 (18.2)	49 (44.6)	3 (2.3)	51 (41.8)	48 (42.1)
	Intermediate	42 (29.8)	30 (18.4)	47 (34.6)	80 (51.9)	43 (40.0)	43 (32.8)	36 (29.5)	44 (38.6)
	Low	29 (20.6)	113 (69.3)	30 (22.0)	46 (29.9)	18 (16.4)	85 (64.9)	35 (28.7)	22 (19.3)
**Previous exposure to dengue virus (Dengue Virus DENV)**									
	Yes	26 (18.4)	32 (19.6)	7 (5.2)	16 (10.4)	9 (8.2)	11 (8.4)	16 (13.1)	3 (2.6)
	No	114 (80.9)	129 (79.2)	129 (94.8)	130 (84.4)	101 (91.8)	119 (90.8)	105 (86.1)	109 (95.6)
	Don’t Know	1 (0.7)	2 (1.2)	0 (0.0)	8 (5.2)	0 (0.0)	1 (0.8)	1 (0.8)	2 (1.8)
**DENV immunization**									
	Yes	15 (10.6)	16 (9.8)	5 (3.7)	9 (5.8)	0 (0.0)	0 (0.0)	0 (0.0)	0 (0.0)
	No	117 (83.0)	131 (80.4)	119 (87.5)	136 (88.3)	110 (100)	131 (100)	119 (97.5)	112 (98.3)
	Don’t Know	9 (6.4)	16 (9.8)	12 (8.8)	9 (5.8)	0 (0.0)	0 (0.0)	3 (2.5)	2 (1.7)
**Yellow Fever immunization**									
	Yes	0 (0.0)	8 (4.9)	1 (0.7)	1 (0.7)	0 (0.0)	0 (0.0)	0 (0.0)	0 (0.0)
	No	131 (92.9)	105 (64.4)	82 (60.3)	138 (89.6)	110 (100)	128 (97.7)	119 (97.5)	112 (98.3)
	Don’t Know	10 (7.1)	50 (30.7)	53 (39.0)	15 (9.7)	0 (0.0)	3 (2.3)	3 (2.5)	2 (1.7)
**Duration of living in the residence (year)**									
	≤5	5 (3.6)	10 (6.1)	9 (6.6)	4 (2.6)	5 (4.5)	9 (6.9)	1 (0.8)	2 (1.8)
	>5	136 (96.4)	153 (93.9)	127 (93.4)	150 (97.4)	105 (95.5)	122 (93.1)	121 (99.2)	112 (98.2)
**Number of people living in household**									
	1–3	57 (40.4)	42 (25.8)	68 (50.0)	59 (38.3)	21 (19.1)	18 (13.7)	23 (18.9)	18 (15.8)
	4–5	60 (42.6)	80 (49.1)	37 (27.2)	77 (50.0)	41 (37.3)	48 (36.7)	51 (41.8)	54 (47.4)
	>5	24 (17.0)	41 (25.1)	31 (22.8)	18 (11.7)	48 (43.6)	65 (49.6)	48 (39.3)	42 (36.8)
**Sleeping under a bed-net (daytime)**									
	Yes	4 (2.8)	4 (2.5)	13 (9.6)	15 (9.7)	9 (8.2)	4 (3.1)	6 (4.9)	9 (7.9)
	No	137 (97.2)	159 (97.5)	123 (90.4)	139 (90.3)	101 (91.8)	127 (96.9)	116 (95.1)	105 (92.1)
**Sleeping under a bed-net (nighttime)**									
	Yes	41 (29.1)	84 (51.5)	95 (69.9)	150 (97.4)	73 (66.4)	103 (78.6)	96 (78.7)	83 (72.8)
	No	100 (70.9)	79 (48.5)	41 (30.1)	4 (2.6)	37 (33.6)	28 (21.4)	26 (21.3)	31 (27.2)
**Any kind of larval mosquito control**									
	Yes	140 (99.3)	152 (93.3)	134 (98.5)	154 (100)	108 (98.2)	95 (72.5)	119 (97.5)	100 (87.7)
	No	1 (0.7)	11 (6.7)	2 (1.5)	0 (0.0)	2 (1.8)	36 (27.5)	3 (2.5)	14 (12.3)
**Any kind of adult mosquito control**									
	Yes	70 (49.6)	86 (52.8)	132 (97.1)	150 (97.4)	103 (93.6)	104 (79.4)	113 (92.6)	100 (87.7)
	No	71 (50.4)	77 (47.2)	4 (2.9)	4 (2.6)	7 (6.4)	27 (2.6)	9 (7.4)	14 (12.3)
**Window screen**									
	Yes	43 (30.5)	4 (2.5)	46 (33.8)	2 (1.3)	17 (15.5)	0 (0.0)	16 (13.1)	10 (8.8)
	No	98 (69.5)	159 (97.5)	90 (66.2)	152 (98.7)	93 (84.5)	131 (100)	106 (86.9)	104 (91.2)

DENV—Dengue Virus.

**Table 2 ijerph-17-09134-t002:** Risk factors association with positive dengue antibody seroprevalence in May 2019 in Thailand and Laos (N = 975).

Variables		N (%)	No. of People Seropositive (% of N)	Univariable Analysis		Multivariable Analysis	
				OR (95% CI)	*p*-Value	Adjusted OR (95% CI)	*p*-Value
**Country**							
	Thailand	503 (51.6)	461 (91.7)	Ref.			
	Laos	472 (48.4)	436 (92.4)	1.10 (0.69–1.76)	0.678	-	-
**Province**							
	Mukdahan	248 (25.4)	226 (91.1)	Ref.		Ref.	
	Ubon Ratchathani	255 (26.2)	235 (92.2)	1.14 (0.61–2.16)	0.677	1.27 (0.45–3.54)	0.653
	Savannakhet	241 (24.7)	215 (89.2)	0.81 (0.44–1.47)	0.484	1.87 (0.71–4.94)	0.210
	Champasak	231 (23.7)	221 (95.7)	2.15 (0.99–4.66)	0.052	3.92 (1.24–12.36)	**0.020**
**Setting**							
	Rural	510 (52.3)	461 (90.4)	Ref.		Ref.	
	Urban	465 (47.7)	436 (93.8)	1.60 (0.99–2.57)	0.055	0.68 (0.28–1.61)	0.375
**Age group (year)**							
	5–9	64 (6.6)	27 (42.2)	Ref.		Ref.	
	10–14	72 (7.4)	43 (59.7)	2.03 (1.02–4.03)	**0.043**	2.32 (0.99–5.45)	0.053
	15–19	42 (4.3)	36 (85.7)	8.22 (3.03–22.30)	**<0.001**	9.10 (2.65–31.30)	**<0.001**
	≥20	797 (81.7)	791 (99.2)	180.7 (70.1–465.8)	**<0.001**	223.0 (68.4–713.4)	**<0.001**
**Gender**							
	Male	323 (33.1)	293 (90.7)	Ref.			
	Female	652 (66.9)	604 (92.6)	1.29 (0.79–2.08)	0.299	-	-
**Socioeconomic status**							
	High	300 (30.8)	284 (94.7)	Ref.		Ref.	
	Intermediate	330 (33.8)	301 (91.2)	0.59 (0.31–1.10)	0.097	1.14 (0.91–1.16)	0.810
	Low	345 (35.4)	312 (90.4)	0.53 (0.29–0.99)	**0.046**	1.23 (0.91–1.32)	0.723
**Previous exposure with DENV**							
	Yes	81 (8.3)	77 (95.1)	Ref.			
	No	882 (90.5)	808 (91.6)	0.57 (0.20–1.59)	0.281	-	-
	Don’t Know	12 (1.2)	12 (100)	Not analyzed		-	-
**Duration of living in the residence (year)**							
	≤5	35 (3.6)	27 (77.1)	Ref.		Ref.	
	>5	940 (96.4)	870 (92.6)	3.68 (1.61–8.42)	**0.002**	1.45 (0.34–6.17)	0.615
**Number of people living in household**							
	1–3	270 (27.7)	256 (94.8)	Ref.		Ref.	
	4–5	409 (41.9)	377 (92.2)	0.64 (0.34–1.23)	0.187	1.89 (0.80–4.44)	0.152
	>5	296 (30.4)	264 (89.2)	0.45 (0.23–0.87)	**0.017**	0.98 (0.53–1.80)	0.940
**Sleeping under a bed net (daytime)**							
	Yes	60 (6.2)	56 (93.3)	Ref.			
	No	915 (93.8)	841 (91.9)	0.81 (0.29–2.30)	0.695	-	-
**Sleeping under a bed net (nighttime)**							
	Yes	675 (69.2)	620 (91.9)	Ref.			
	No	300 (30.8)	277 (92.3)	1.07 (0.64–1.78)	0.798	-	-
**Any kind of larval mosquito control**							
	Yes	907 (93.0)	840 (92.6)	Ref.		Ref.	
	No	68 (7.0)	57 (83.8)	0.41 (0.21–0.83)	**0.013**	0.51 (0.14–1.88)	0.311
**Any kind of adult mosquito control**							
	Yes	792 (81.2)	728 (91.9)	Ref.			
	No	183 (18.8)	169 (92.3)	1.06 (0.58–1.94)	0.847	-	-
**Window screen**							
	Yes	121 (12.4)	118 (97.5)	Ref.		Ref.	
	No	854 (87.6)	779 (91.2)	0.26 (0.08–0.85)	**0.026**	0.44 (0.08–2.44)	0.348

OR—Odds Ratio; CI—Confidence Intervals. The bold numbers indicate that statistical significance was observed.

**Table 3 ijerph-17-09134-t003:** Risk factors associated with dengue antibody seroconversion between May and November 2019 in Thailand and Laos (N = 1071).

Variables		N (%)	No. of People Seroconversion (% of N)	Univariable Analysis		Multivariable Analysis	
				OR (95% CI)	*p*-Value	Adjusted OR (95% CI)	*p*-Value
**Country**							
	Thailand	594 (55.5)	7 (1.2)	Ref.		Ref.	
	Laos	477 (44.5)	26 (5.5)	5.50 (2.19–13.78)	**<0.001**	2.15 (0.29–15.92)	0.450
**Province**							
	Mukdahan	304 (28.4)	4 (1.3)	Ref.		Ref.	
	Ubon Ratchathani	290 (27.1)	3 (1.0)	0.78 (0.17–3.70)	0.760	0.45 (0.03–6.46)	0.551
	Savannakhet	241 (22.5)	10 (4.2)	3.34 (0.91–12.24)	0.069	4.99 (0.61–40.82)	0.134
	Champasak	236 (22.0)	16 (6.8)	6.79 (1.98–23.35)	**0.002**	39.89 (5.34–299.1)	**<0.001**
**Setting**							
	Rural	562 (52.5)	21 (3.7)	Ref.		Ref.	
	Urban	509 (47.5)	12 (2.4)	0.55 (0.24–1.27)	0.161	0.73 (0.18–3.02)	0.668
**Age group (year)**							
	5–9	68 (6.3)	9 (13.2)	Ref.		Ref.	
	10–14	88 (8.2)	13 (14.8)	4.80 (2.37–9.70)	**<0.001**	5.27 (2.34–11.89)	**<0.001**
	15–19	52 (4.9)	2 (3.9)	0.25 (0.10–0.61)	**<0.001**	0.14 (0.05–0.37)	**<0.001**
	≥20	863 (80.6)	9 (1.0)	0.06 (0.04–0.10)	**<0.001**	0.03 (0.02–0.06)	**<0.001**
**Gender**							
	Male	361 (33.7)	18 (5.0)	Ref.		Ref.	
	Female	710 (66.3)	15 (2.1)	0.19 (0.13–0.26)	**<0.001**	1.89 (1.14–3.11)	**0.013**
**Socioeconomic Status**							
	High	328 (30.6)	11 (3.4)	Ref.			
	Intermediate	365 (34.1)	13 (3.6)	1.04 (0.39–2.74)	0.939	-	-
	Low	378 (35.3)	9 (2.4)	0.69 (0.25–1.91)	0.474	-	-
**Yellow Fever immunization**							
	Yes	45 (4.2)	0 (0.0)	Ref.			
	No	975 (91.0)	28 (3.0)	149 (4.72–4.70)	0.710	-	-
	Don’t Know	51 (4.8)	5 (3.7)	1	-	-	-
**Duration of living in the residence (year)**							
	≤5	45 (4.2)	2 (4.4)	Ref.		Ref.	
	>5	1026 (95.8)	31 (3.0)	0.04 (0.02–0.10)	**<0.001**	0.005 (0.001–0.02)	**<0.001**
**Sleeping under a bed net (daytime)**							
	Yes	64 (6.0)	0 (0.0)	Ref.		Ref.	
	No	1007 (94.0)	33 (3.3)	20.74 (0.61–706.27)	0.092	8.55 (0.44–166.57)	0.158
**Sleeping under a bed net (nighttime)**							
	Yes	725 (67.7)	21 (2.9)	Ref.			
	No	346 (32.3)	12 (3.5)	1.25 (0.54–2.92)	0.600	-	-
**Any kind of larval mosquito control**							
	Yes	1002 (93.6)	28 (2.8)	Ref.		Ref.	
	No	69 (6.4)	5 (7.3)	3.91 (1.10–13.89)	**0.040**	3.38 (0.52–22.12)	0.210
**Any kind of adult mosquito control**							
	Yes	858 (80.1)	27 (3.2)	Ref.			
	No	213 (19.9)	6 (2.8)	0.99 (0.36–2.74)	0.990	-	-
**Window screen**							
	Yes	138 (12.9)	3 (2.2)	Ref.			
	No	933 (87.1)	30 (3.2)	1.92 (0.46–8.02)	0.370	-	-

The bold numbers indicate that statistical significance was observed.

**Table 4 ijerph-17-09134-t004:** Mean monthly values (and maxima and minima) of temperature (°C), relative humidity (%) and cumulative rainfall (mm) for the four provinces. DTR—Diurnal temperature range.

Province	Mean Monthly (Range) from May to November 2019					
	Max Temp	Min Temp	Mean Temp	DTR	Humidity	Cumulative Rainfall
**Mukdahan**	28.5 (25.9–30.6)	26.7 (23.8–28.9)	27.6 (24.8–29.7)	1.73 (1.28–2.09)	81.0 (74.5–88.8)	323.9 (4–1229)
**Ubon Ratchathani**	28.4 (25.9–30.6)	26.7 (23.9–29.1)	27.5 (24.9–29.9)	1.67 (1.32–2.00)	82.9 (77.1–88.7)	150.2 (0–398.8)
**Savannakhet**	28.7 (25.7–31.2)	27.1 (23.6–29.7)	27.9 (24.6–30.4)	1.63 (1.18–2.09)	78.5 (70.6–88.1)	168.1 (0–562.4)
**Champasak**	29.3 (27.3–35.3)	27.7 (25.4–33.2)	28.5 (26.4–34.2)	1.53 (1.19–2.08)	80.8 (72.5–87.9)	254.5 (0.64–676.6)

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
