# Peer review of "Dengue Seroprevalence and Seroconversion in Urban and Rural Populations in Northeastern Thailand and Southern Laos"

_ijerph, 2020, doi:10.3390/ijerph17239134_

Round 1

Reviewer 1 Report

The manuscript "Dengue seroprevalence and seroconversion in urban and rural populations in northeastern Thailand and southern Laos" utilizes an expansive study population, with well-designed criteria to answer an interesting question about the overall seroprevalence in this region of southeast Asia. Overall the manuscript could benefit from some basic English grammar and writing proofing.  

Major issues 

-Overall, the universally high seroprevalence poses a substantial problem for the proper analysis. 

-The youngest cohort of the <5-year-olds is significantly higher in the Rural populations.  This is something that should have been addressed at enrollment and once again should be stated.  Especially since these subjects are the least likely to be seropositive.

-As a third critique of the participant selection, males are almost twice as numerous as females.

            All of the above critiques may not be able to be addressed as the study is likely over. I know there is nothing that can be done in hindsight, but I would like the authors to qualify some of their statements with these facts, possibly in the discussion.

-The participants that received the Dengue vaccine in Thailand should be excluded.  As this paper is focusing on a immune readout of infection.  I believe that they can remain in the manuscript, but as you do not know if they are positive from the vaccine or an actual infection, they should be removed from the analysis.  I cannot find mention of their exclusion.

-The numbers for the seroconversion of the participants in Champasak province does not make sense to me as well.  Originally it was stated that of the 236 people 225 were seropositive, but then 16 seroconverted.  Which does not add up.  If I missed something in the text I apologize, but then that should be made clearer.  It seems like the back half of the paper is weighted on this fact.  If the seroconversion data for this region was misinterpreted, you will have to make subsequent changes to the back portion of the manuscript.

Minor Adjustments

-Figure one- please move regional names in boxes from the border of the overall figure.  The 2 borders look incorrect when they are on top of each other.

-The table headers should be made in a way to not break up worlds onto multiple lines.  IE the regional names in table one.  Please reformat the figure to avoid this.  It happens in all of the attached tables.

-Please elaborate on the portion that looks at previous exposure to DENV.  Is that just a questionnaire answer?

-Please increase the size of the units on both the X and Y axis of  fig 2, and change the regression line to black to differentiate it from the positive and negative data sets.

-Overall, the paper could use some more proofreading.  Such as started a sentence with an actual number like in line 46.

Reviewer 2 Report

The paper by Dyna Doum entitled dengue seroprevalence and seroconversion in urban and rural populations in northeastern Thailand and southeast Laos. was reviewed.

The paper was well conducted survey on dengue antibody in Thai and Laos. Dengue infection is one of the most important tropical infectious diseases.  Studies on seroprevalance is important because it is known that reinfection caused more severe forms.  The study here analyzed many important environmental and living style which might affect the seroprevelance.

This works is solely dependent on commercial ELISA. No objective evaluation has been made on the performance of this test reagent. They should provide the comparative data using another kit and non-endemic control blood samples. It is not clear which population and   how often paired samples were taken.  It might be useful to show the data of clinical severity of seropositive individuals when they developed DENV symptoms.

Round 2

Reviewer 2 Report

In the newly cited paper , "Comparison of six commercial diagnostic test ......" , it was not described which samples (plasma or serum) were used except some serum samples from Korea. Plasma and serum should have different C.O.I. I am not convinced how the authors separated serum from DTA tubes. More careful criteria for plasma analysis should be given.

Author Response

Reviewer's comment:

In the newly cited paper , "Comparison of six commercial diagnostic test ......" , it was not described which samples (plasma or serum) were used except some serum samples from Korea. Plasma and serum should have different C.O.I. I am not convinced how the authors separated serum from DTA tubes. More careful criteria for plasma analysis should be given.

Answer

  • The plasma or serum samples do not have different COI, according to the previous study comparing the performance of serum and plasma by Stuart D. Blacksell et al. (2012) https://rp.liu233w.com:443/https/www.ncbi.nlm.nih.gov/pmc/articles/PMC3435366/pdf/tropmed-87-573.pdf, where they found serum and plasma samples were not significantly different.
  • The ELISA kit (EUROIMMUN) that we used can be performed with human serum or EDTA or heparin plasma.
  • We replaced the previous reference (cited above by the reviewer) with a new very recent reference on sensitivity and specificity (ref. 17) where they tested two ELISA kits (including the one we used) and three RDTs (line 127).

DiazGranados, C.A., et al., Accuracy and efficacy of pre-dengue vaccination screening for previous dengue infection with five commercially available immunoassays: a retrospective analysis of phase 3 efficacy trials. The Lancet Infectious Diseases, 2020.

They selected five immunoassays assessed, two ELISAs (EUROIMMUN and Panbio), and three rapid diagnostic tests (RDTs; TELL ME FAST, SD BIOLINE, and OnSite) to test baseline samples by IgG-based immunoassays to classify baseline dengue serostatus. The reason they selected these kits was because these assays had shown high specificity and low cross-reactivity.